# COVID-19 Vaccination Intentions amongst Healthcare Workers: A Scoping Review

**DOI:** 10.3390/ijerph191610192

**Published:** 2022-08-17

**Authors:** Lucia D. Willems, Vernandi Dyzel, Paula S. Sterkenburg

**Affiliations:** 1Faculty of Behavioural and Movement Sciences, Vrije Universiteit Amsterdam, 1081 BT Amsterdam, The Netherlands; 2Moments Psychology, 1016 GL Amsterdam, The Netherlands; 3Bartiméus, 3941 XM Doorn, The Netherlands

**Keywords:** COVID-19, health personnel, intellectual disability, vaccination attitudes, vaccination hesitancy, vaccination intention

## Abstract

A worldwide vaccination programme is the chosen strategy against the COVID-19 pandemic. Vaccine hesitancy, however, forms a threat to achieving a high degree of vaccination. Healthcare workers (HCWs) are exposed to greater risks, in addition to HCWs who care for people with intellectual disabilities (ID). However, little is still known about these groups’ vaccine hesitancy. This review aims to provide insight into the intentions and attitudes of HCWs on COVID-19 vaccination, including those who care for people with ID. The search included both types and was conducted in nine databases. A total of 26 papers were identified concerning the vaccine intentions of 43,199 HCWs worldwide. The data were gathered both quantitively and qualitatively. The papers were analysed for all of the themes regarding vaccine intentions, which were: (1) percentages of vaccine willingness; (2) predictors of willingness; (3) attitudes of willingness and hesitancy; (4) sources of vaccination information; (5) contextual factors and changes in COVID-19 vaccine acceptance over time; and (6) future strategies for interventions. Concerns about vaccine safety, efficacy and short- and long-term side effects were the most prominent in HCWs and, therefore, should be addressed in future intervention strategies. Furthermore, interactive interventions are recommended to facilitate exchange, and accurate information should be accessible to target groups on social media platforms.

## 1. Introduction

The COVID-19 vaccination campaigns started worldwide in December 2020 and are the primary strategy to tackle the pandemic. Along with vulnerable population groups, such as the elderly and individuals with vulnerable health, healthcare workers (HCWs) were prioritised to be vaccinated [1,2]. One reason was the heightened occupational risk that all HCWs are exposed to because they care for vulnerable population groups [3], who are more prone to COVID-19 infections. Consequently, HCWs are at a greater risk of direct exposure to the virus. The other reasons to prioritise HCWs are to prevent them from infecting the patients they care for [4] and to keep HCWs fit to work where they are most needed as frontline workers [3].

When the first vaccines for COVID-19 were approved in December 2020, the strategy was to achieve group immunity [5], which would be achieved when 67% of the population was vaccinated [6]. More recent evidence [7] states that classic group immunity for COVID-19 is an impossible goal, due to new and upcoming virus variants [8], asymptomatic transmission and because immunity is no longer guaranteed after infection or vaccination [9]. Thus, the best way forward is to achieve the highest vaccination degree possible instead of aiming at a threshold value.

Since vaccine uptake is generally voluntary, vaccine willingness plays a vital role in a successful vaccination campaign [10]. The vaccine intentions of individuals are often influenced by attitudes, beliefs and opinions on vaccination [11], and are expressed in either vaccine willingness or vaccine hesitancy. Therefore, it is essential to investigate the reasons and arguments for vaccine uptake or refusal. Several studies have investigated the factors influencing vaccine willingness and vaccine hesitancy, such as the beliefs and attitudes towards vaccination. A global survey in 19 countries measured potential vaccine acceptance in the general population [12] and found rates of 88.62% in China, compared to 54.85% in Russia. The extensive systematic review on vaccine willingness by Sallam [13] reported on a total of 60 surveys taken from 31 published papers, including the survey of Lazarus and colleagues [12], but did not describe the attitudes associated with vaccine willingness or hesitancy. The included papers were carried out globally in the general population and different target population groups, such as HCWs, students and parents. The surveys analysing the general population (*n*
*=* 47) found the highest vaccine willingness rates in Ecuador (97%) and the lowest in Kuwait (23.6%). The surveys analysing HCWs (*n*
*=* 8) found the highest vaccine willingness in Israel (78.0%) and the lowest in the Democratic Republic of the Congo (DRC, 27.7%). Hence, a review study examining the arguments for vaccine willingness and hesitancy of HCWs can provide important information for policymakers to reach the highest vaccination degree possible among HCWs.

### 1.1. HCWs in the Field of Intellectual Disability

HCWs who care for people with intellectual disabilities (ID) face a heightened risk due to the group they attend and the conditions in which this group lives. The people with ID are vulnerable as they often have multiple additional medical comorbidities [14]. Furthermore, people with ID have a higher chance of infection [15], as they often stay in assisted-living facilities throughout their lives, where they, depending on the severity of their disability, are cared for by a large number of HCWs. The higher the number of individuals with whom the patients are in contact, the higher the risk for infection. Indeed, several studies have reported excess mortality rates during the pandemic among people with ID [16,17,18]. In the Netherlands, for example, the Academic Collaborative ‘Sterker op eigen Benen’ found that, while 1.1% of the general population who contracted a COVID-19 infection have died, the mortality figure of 4.6% for people with ID was far higher [19]. Thus, the HCWs who care for people with ID could face a higher risk of contracting COVID-19 than other HCWs. Therefore, it is crucial to investigate the reasons and arguments for vaccine uptake or refusal of the HCWs caring for people with ID, given their heightened risk.

### 1.2. The Current Paper

To date, however, very few studies have explored the vaccine willingness and hesitancy of HCWs in general and, more specifically, of the HCWs employed in the care for people with ID and their attitudes towards COVID-19 vaccination. This review aims to better understand vaccine willingness and hesitancy among HCWs in general, and those who care for people with ID, by identifying key themes in the available research. The research question is: What factors are associated with vaccination hesitancy and willingness in HCWs? The current study assesses this question by conducting a scoping review: to synthesize the available topics and identify gaps in the research field [20]. Thus, all of the themes related to vaccine willingness and hesitancy were gathered from the reviewed articles.

## 2. Materials and Methods

### 2.1. Design and Protocol

This scoping review aims to identify the key themes in the literature concerning vaccination hesitancy and willingness among HCWs. In order to attain high-quality reporting, the Preferred Reporting Items for Systemic reviews and Meta-Analyses extension for Scoping Reviews (PRISMA-ScR) Checklist was used in this review [21]. The 12-step guideline by Kable et al. [22] was used during the search strategy.

### 2.2. Eligibility Criteria

The papers that focused on COVID-19 vaccination willingness and hesitancy amongst HCWs were included. Initially, the search focused on those HCWs who cared for people with ID, but very few papers met this criterium. Therefore, the scope was broadened to include HCWs in general. The pre-printed and published, qualitative, quantitative and commentary papers were included. Because this is a scoping review, the papers that were not original research were also included, such as a systematic review and a commentary. There was no language exclusion. There was no eligibility criterium for the year of publication or data collection timeframe, but only the papers dating from 2020 and 2021 were available due to the search topic. The papers that did not focus on vaccination hesitancy or willingness amongst HCWs and their corresponding attitudes, beliefs, perceptions or views were excluded.

### 2.3. Information Sources, Search Strategy and Selection Process

The second author conducted the search on 4 April 2021, in the databases CINAHL, APA PsycArticles and APA PsycInfo (via the EBSCO host), Web of Science, Semantic Scholar, Prospero, Outbreak Science, Cochrane and Scopus. The search terms were variants of the keywords COVID-19, vaccination intentions, healthcare workers and intellectual disability, combined with Boolean search operators. A complete description can be found in the Appendix A. This resulted in the identification of articles related to HCW perceptions. Figure 1 displays all of the steps conducted to reach a total of 26 articles.

### 2.4. Data Synthesis and Analysis

Microsoft Excel was used for the following steps. First, the purpose of each of the studies was extracted from the introduction texts. If present, the hypotheses were also extracted. Second, the methodology of each of the studies was extracted, including the type of research (cross-sectional survey/qualitative report/systematic review) and the survey topics (e.g., sociodemographic characteristics, vaccination intent, reasons for vaccine hesitancy, etc.). Third, the samples of the studies were described: sample sizes, type of participants/respondents (HCWs, general population, students), group sizes and descriptors of participants (e.g., gender, profession, age, level of education). Fourth, the countries and time frame of the data collection were extracted. The time frame of the data collection is essential in this case, because vaccination willingness/hesitancy may change throughout the pandemic. Finally, the studies’ results and conclusions were gathered and organised into themes and sub-themes. These themes are (1) *Percentages of vaccine willingness;* (2) *Predictors of vaccine willingness* differentiated by 11 sub-themes; (3) *Attitudes of vaccine willingness and hesitancy* differentiated by 19 sub-themes; and (4) *Sources of vaccination information*. Two more themes that will be discussed directly in the discussion section (Section 4), of this review are (5) *Contextual factors and changes in COVID-19 vaccine acceptance over time*; and (6) *Future strategies for interventions*. The difference between themes two and three is that theme two entails the socio-demographical characteristics of the respondents: attributes that are descriptors, while theme three entails the thoughts, beliefs, or opinions of the respondents: attributes that are susceptible to change.

After the second author searched, the first author independently reviewed all of the 27 papers in the final selection for eligibility by making a full-text screening. The paper by Nioi and colleagues [23] was omitted from the final selection after consultation with all of the authors, resulting in 26 papers. Although this paper [23] does cover future challenges in the vaccination campaign specifically toward HCWs, it does not cover vaccine hesitancy or willingness amongst HCWs and their attitudes on the subject. Because Sallam [13] is a systematic review, their results will be compared to those of the current review in the discussion section (Section 4), of this paper. The commentary by Gur-Arie et al. [24] discusses vaccine hesitancy among HCWs and the ethical issues of mandatory COVID-19 vaccination and will also be reviewed in the discussion section (Section 4).

Next, all of the themes extracted from the papers by the second author were reviewed by the first author. All of the authors came to a consensus about the themes identified. There was one alteration of a category: *Contextual factors* now also includes *Changes in COVID-19 vaccine acceptance over time*. Two of the attitudes were merged, and one theme was rearranged to make a more concise overview of themes.

### 2.5. Characteristics of the Literature

The characteristics of the papers included in this review are presented in Table 1. In the running text of Section 2 and Section 3, the referencing numbers of the reviewed papers are marked in ^superscript^ and correspond with the paper numbers 1–26 in all of the tables. The screening for the papers on the topic of COVID-19 vaccination willingness and hesitancy amongst HCWs yielded 26 articles that mainly consisted of cross-sectional surveys (*n* = 22). The remaining four papers were two cross-sectional qualitative reports^1,14^, a commentary^7^ and a systematic review^21^. The data contained in the included papers were gathered in 16 different countries worldwide, spread out over the continents of Africa, Asia, Europe, North America or a combination of two continents. The time frame for the data collection of these papers was between February 2020 and March 2021. The sample sizes ranged from 24 respondents in a cross-sectional qualitative report^14^ to 16,158 respondents in a cross-sectional survey^15^. In 12 papers^2,3,4,5,9,11,17,18,19,22,24,26^ the group of HCWs was differentiated by health professions within the health care system, as seen in the *Respondents* column. Their corresponding vaccine willingness is also shown in Table 1.

## 3. Results

### 3.1. The Difference in Vaccine Willingness across Samples

A total of 21 papers reported percentages representing vaccine willingness amongst HCWs. These percentages differed drastically across the papers, ranging from only 27.7% in one paper^9^ to 92% in another^18^. Vaccine willingness among HCWs is lowest in the Democratic Republic of the Congo (27.7%)^9^, Africa, followed by three countries in the Middle East: Egypt (45.9%)^16^, Saudi Arabia (50.52%)^20^ and Turkey (52.9%)^26^. Only one paper reported on the vaccine willingness of an Asian country: China (63%)^12^. The vaccine willingness of HCWs in North America fluctuated the most, with vaccine willingness ranging from 36%^22^ to 92%^18^, both in the USA. In Europe, vaccine willingness amongst HCWs ranged from 56%^6^ in France to 82.95%^23^ in Poland. The 13 papers that differentiated vaccine willingness across various health professions will be further explained in the following section *Predictors of Vaccine Willingness*.

Only two papers^8,13^ reported about HCWs employed in the care of people with ID. In both of the papers, HCWs showed a comparable percentage of vaccination intention; 76% and 82%. These papers collected data between January and February 2021, when vaccination was not mandated in the USA and Canada. Iadarola and colleagues [30] focused not only on the HCWs who care for people with ID but also individuals diagnosed with ID and family members of people with ID (e.g., individuals who spend time with people with ID and make decisions for them). The vaccine intentions did not significantly vary across these three groups.

Not all of the papers reported exclusively on the vaccine willingness of HCWs but also included the willingness of other populations^3,4,8,21,24,26^. Of the papers that reported about HCWs compared to the general population, some mentioned significantly higher vaccination willingness percentages for HCWs than the general population^23,26^. In contrast, others found no statistically significant difference between these groups^4,8^, and one found that doctors and the general population were more willing to get vaccinated than nurses^3^. In an additional paper^23^, the vaccination willingness of medical students (i.e., future HCWs) was compared to that of non-medical students, with significantly higher percentages for the medical students.

### 3.2. Predictors of Vaccine Willingness

The theme *Predictors of Vaccine Willingness* was split into 13 sub-themes, presented in Table 2. Not all of the sub-themes are statistically significant predictors in each paper. *Profession*, *age*, *past vaccine behaviour* and *gender* are statistically significant predictors in at least eight papers. The sub-themes that were significant predictors in less than five papers were *comorbidities*, *education*, *ethnicity*, *mental well-being*, *COVID self-history*, *geographical location*, *COVID family history*, *income* and *political orientation*.

The type of profession was questioned in 18 papers and significantly predicted vaccine willingness in 12 papers^2,3,4,5,9,11,17,18,19,22,24,26^. Physicians or medical doctors had higher vaccine willingness than other HCW staff in 11 of those papers. Szmyd and colleagues [45] reported vaccine willingness as high as 94.44% among physicians (MD), whereas only 61.48% of administrative healthcare assistants (HA) were willing to get vaccinated. One paper^11^ differed from the rest of the results: physicians were associated with lower vaccine willingness than nurses, other HCWs and students, as shown in Table 1.

Age also significantly predicted vaccine willingness. Age was questioned in 21 papers, and was a statistically significant predictor in ten papers^2,8,9,11,12,13,16,18,22,26^. Older age predicted a higher vaccine willingness in most papers^8,9,13,16,22,26^, but not all of them. Three papers^2,11,12^ reported a younger age as predicting vaccine willingness. Finally, vaccination intention was highest in the age groups under 40 and over 60 in one paper^18^.

Both past vaccine behaviour and gender predicted vaccine willingness frequently and consistently. A total of 10 papers^2,3,5,11,18,20,22,23,24,25^ out of 13 reported that past influenza vaccine behaviour predicted vaccine willingness significantly; receiving the influenza vaccine corresponded with higher vaccine intentions. A total of eight papers^3,5,9,11,18,20,22,26^ out of 21 reported that gender predicted vaccine willingness; being male corresponded with higher vaccine intentions.

### 3.3. Attitudes on Vaccine Willingness and Hesitancy

Eighteen different sub-themes focusing on the attitudes related to vaccine willingness and hesitancy were extracted from the 26 selected papers. As shown in Table 3, all of the attitudes corresponded either with vaccine willingness or hesitancy and were categorised into five overarching themes: *Health factors*; *Ethics*; *Trust-related issues*; *Information* and *Practical factors*. Where possible, the information about significance was added. However, most of the papers only reported significant relations between attitudes and vaccine willingness. The attitudes corresponding to vaccine hesitancy were primarily ranked in order of frequency. Therefore, the attitudes mentioned most frequently concerning vaccine hesitancy were also marked.

The attitudes or beliefs that consistently corresponded with vaccine willingness fall under the themes of *Health factors* and *Ethics*. The most frequently occurring attitude in favour of vaccine willingness was the perceived COVID-19 threat: fifteen different papers reported that an increased perception of COVID-19 threat was associated with higher vaccine willingness. Six papers reported that attitudes toward protecting family members, vaccination being part of the job and collective responsibility were associated with vaccine willingness. Finally, the belief that vaccination should be mandatory was measured and linked to vaccine willingness in three papers. Here, HCWs were more willing to get vaccinated than other groups of participants and thought more often that vaccination should be mandatory.

The attitudes or beliefs that consistently correspond with vaccine hesitancy can also be seen in Table 3. Eight of the 13 attitudes that corresponded with hesitancy fall under the theme of *Trust-related issues*, one under the theme *Information* and four under the theme *Practical factors*. The attitude associated with hesitancy most often was fear of short or long-term side effects; this was the case in 17 papers. The additional concerns reported in ten or more papers that fall under the category *Trust-related issues* were the vaccine’s safety, efficacy, and speed of development. Distrust was observed in several forms: distrust of the vaccine itself; distrust of the government; distrust of the pharmaceutical companies or distrust of health authorities/officials. Another attitude associated with vaccine hesitancy was the lack of information or misinformation; this was the case in nine papers. Examples of where information was lacking were: (1) the vaccine in general^14^; (2) the benefits of vaccination^26^; (3) vaccine development^2,18^; (4) the financial interest of pharmaceutical companies^11^ and (5) vaccine side effects^11^. The spread of misinformation across traditional and social media was established in two qualitative papers^1,14^ and four quantitative papers^9,16,24,26^. Finally, four attitudes that corresponded with vaccine hesitancy fall under the theme of *Practical factors*. The attitude of favouring alternative (herbal or organic) treatments over vaccination was linked to hesitancy in three papers. The impact on other precautions, logistics to get vaccinated and spirituality/religion were all linked to hesitancy once.

### 3.4. Information Sources of COVID-19 and Vaccination

In some of the papers, vaccination distrust and vaccine hesitancy were linked to the sources used by the respondents for the COVID-19 vaccination information. For instance, Di Gennaro and colleagues [26] reported that using Facebook as a primary source of information on the COVID-19 vaccination was a statistically significant predictor of higher vaccine hesitancy. In the paper by Mohamed Hussein and colleagues [38], the prime sources of information for respondents were social media (77%) and television (40%), and according to the authors, this was a reason for the low percentage of vaccine willingness in their sample, which was 45.9%. Additionally, a positive relation between the vaccine hesitancy and distrusting the Ministry of Health was found by Verger and colleagues [46]. They reasoned that this distrust was a major problem since the Ministry of Health was the primary information source on COVID-19 vaccination, for both HCWs and the general population.

Other sources of information on COVID-19 than Facebook or social media were linked to vaccine willingness. Attending lectures/discussions about COVID-19 and using official websites were, amongst other variables, significantly associated with COVID-19 vaccine acceptance in the paper by Kabamba and colleagues [31]. Similarly, Yurttas and colleagues [47] describe that, although social media and television were the most common information sources in both the general population and HCWs, the HCWs also frequently consulted institutional declarations and medical literature, which might have caused their higher vaccine acceptance.

## 4. Discussion

This scoping review aimed to examine the key themes in the available research on COVID-19 vaccine willingness and hesitancy in HCWs, including those who care for people with ID, with a specific interest in their arguments, beliefs and attitudes on vaccination. A total of 26 papers were selected and analysed, focusing on the socio-demographical predictors of vaccine willingness and the attitudes that inspire vaccine hesitancy in HCWs. Only two papers focussed on HCWs working with individuals with ID [30,35]. The vaccine willingness percentages were comparable between the respondents of these two papers and within the subgroups of these papers.

As a result of the limited literature available on the HCWs that work with people with ID, the scope of this review was broadened to all HCWs. Therefore, in the following steps, the analyses included HCWs of any kind. First, the predictors of vaccine willingness in all of the 26 papers were analysed. This analysis revealed that profession, age, gender and past vaccine behaviour were the most powerful predictors of the willingness to be vaccinated: medical doctors; people of older age; men; and those previously vaccinated with the influenza vaccine were most likely to receive the COVID-vaccine. Our findings suggest that nursing staff, people of younger age, women and those not previously vaccinated are the target populations when aiming to inform HCWs of the benefits of vaccination. The results of this review partly support those found in the rapid systematic review by Robinson et al. [48]: people of younger age and women were more hesitant towards vaccination in the general population. Additionally, Robinson and colleagues [48] reported higher vaccination hesitancy in people with a lower income, people with lower education and people belonging to an ethnic minority group. These last three respondent groups were statistically significant predictors in only one (income) or two (education and ethnicity) papers in this review.

Next, all of the COVID vaccine attitudes were categorized: first in theme; secondly according to whether they contributed to either vaccine willingness or vaccine hesitancy. The attitudes promoting vaccine willingness fit within the themes of *Health factors* and *Ethics*. Some of the examples are the COVID-19 threat to oneself or the desire to protect family members or other persons. The attitudes promoting vaccine hesitancy fit within the *Trust-related issues* and *Information* themes. The examples of trust-related issues are concerns about both the short- and long-term side effects of the vaccine, the vaccine’s safety, efficacy and the speed of the vaccine development. Other examples are lack of information or misinformation on the vaccine. This suggests that the themes that need to be covered when providing interventions to inform HCWs about vaccination are *Trust-related issues* and *Information*. The findings of this review complement those of earlier studies. Robertson et al. [49] reported, in the UK Household Longitudinal Study with over 12,000 participants, that the main reason for vaccine willingness is reducing the risk of being infected by COVID-19 or getting ill from it (54.6%), while the main impediment to vaccination is worrying about the unknown future effects of the COVID-19 vaccine (42.7%). Interestingly, this paper also reported responses about persuasion. The factors that would increase the likelihood of getting vaccinated, according to the participants, were knowing a vaccine reduced their risk of COVID-19 infection (67.8%), knowing a vaccine reduced the risk of being seriously ill (63.7%) and knowing the vaccine was proven to be safe (59.8%). In other words, the respondents expressed the need for more information on the vaccines’ safety and efficacy.

In the two papers that describe vaccine hesitancy in HCWs that care for people with ID [30,35], the main socio-demographical predictor of willingness was age, with younger people being more hesitant for vaccination. The other socio-demographical characteristics, such as profession, gender, ethnicity, education and geographical location, were not statistically significant predictors of vaccine willingness. In Iadarola et al. [30], there was, however, an interaction effect between age and ethnicity. In both Black and White respondents, those over the age of 50 were less vaccine-hesitant than their younger counterparts. This age difference was less great in the Asian and Latin/Hispanic respondents. The respondents who were hesitant to get vaccinated in the study of Iadarola et al. [21] were most concerned by the vaccine’s possible side effects and its fast development. Not wanting to be an experiment and not trusting the government were the follow-up reasons for vaccine hesitancy, and these opinions were strongest among the Black respondents. The respondents that were hesitant to get vaccinated in the study of Lunsky et al. [35] believed more strongly than those willing to get vaccinated that ‘a vaccine was unnecessary because of good health’, but also distrusted the vaccine because of its fast development and potential side effects. Alternatively, these respondents believed less strongly than those willing to be vaccinated that they were at risk of getting COVID, that a vaccine would protect their family or clients, or that vaccination was a part of their job. Comparing the outcome measures of these two papers reveals the similarities between the respondents: age being the sole predictor of vaccine willingness, and concerns about the vaccines’ side effects and their fast development inspiring vaccine hesitancy.

Finally, the information sources used about the COVID-19 vaccination and their connection to vaccine willingness and hesitancy were analysed. What sources are used to inform oneself on COVID-19 vaccination can influence deciding whether to get vaccinated or not. It seems that some sources, such as Facebook and other forms of social media, are a breeding ground for misinformation and even conspiracy theories, which in turn could lead to vaccine hesitancy. Misinformation is often the basis on which distrust of the vaccine is formed. Berry and colleagues [25] explicitly state: “During town hall meetings with 196 frontline staff from the skilled nursing facility workforce, misinformation through social media was common: rapid vaccine development and infertility and pregnancy-related concerns were among the most frequent raised.” They even state that the concerns raised are more pervasive than some national polls might suggest. Thus, the spread of misinformation through social media platforms, in both the general population and HCWs, should be taken very seriously and counterbalanced by targeted interventions, such as information sessions on actual vaccine information and the spread of more informative videos on social media. Even though social media is the primary source of misinformation, this platform can also be used to make more accurate information, in easy-to-understand language, accessible to our target group and the general public.

### 4.1. Contextual Factors and Changes in COVID-19 Vaccine Acceptance over Time

It is important to note that vaccine intentions may not be stable over time. Some papers collected data on the opinions of HCWs before the distribution of vaccines worldwide in December 2020. Other papers started collecting data after the onset of the mass vaccination programme. The contextual factors could have affected the vaccine willingness of respondents. For example, O’Brien and colleagues [39] collected data from HCWs in the USA using a brief survey sent out in October 2020 (*n*
*=* 2070) and again in December 2020 (*n*
*=* 1541). The vaccine willingness of respondents increased by more than 20% between these two measurements. This increase is partly due to the 998 participants that contributed to both of the measurements and changed their vaccine intention between these two time points: 2.9% changed from willing to hesitant, and 13% from hesitant to willing. Two events may have affected the respondents’ vaccine willingness during these months. The first event was the US presidential election on 3 November 2020 [50,51]. The second event was the release of the phase III clinical trial results by two vaccine manufacturers, after which both of the vaccines were approved for Emergency Use Authorization (EUA) [52]. Similar results were found by Meyer and colleagues [37]: an increase in the daily-recorded cases of vaccine willingness was reported after the EUA release.

Besides the rise of vaccination intentions over time, it is also possible that these intentions could decline. In Qattan et al. [42], the low rates were explained by a decline in daily recorded COVID-19 infections, resulting in alleviated worries. Similarly, Kwok and colleagues [34] speculated that their participants’ relatively low vaccination intentions were “in part explained by the successful crisis response on the COVID-19 epidemic in Hong Kong”. In other words, there might be a lower feeling of urgency for vaccination among these HCWs, if the alternative measures combatted the pandemic well. These results suggest that examining the population’s vaccine willingness and hesitancy is ongoing, at least as long as the COVID-19 pandemic has not ended. In addition, when running vaccine information campaigns, contextual factors should be considered—e.g., new virus variants, research related to this virus and ‘booster’ vaccine campaigns.

### 4.2. Future Strategies for Interventions

Two papers included in this review explicitly described interventions that were held to inform HCWs about COVID-19 vaccination. First, online town hall meetings were organized by Berry and colleagues [25], where moderators and doctors met with the skilled nursing facility (SNF) workforce to discuss COVID-19 vaccination. The qualitative report sheds light on the concerns and questions of 193 HCWs on vaccination and presents examples of the suggested answers by the moderators and doctors. What is interesting about their approach is that these answers were personal stories and experiences whenever possible, instead of data-focused answers. Berry and colleagues [25] conclude that “sharing positive emotions and stories may be more effective than sharing data when attempting to reduce vaccine hesitancy in SNF staff”. It may well be possible that these stories are more relatable than statistics. This should also be taken into account in future campaigns. Interventions should provide information so that the target group can relate to it and make the idea of getting a vaccine less clinical. Second, Gakuba and colleagues [29] investigated the impact of a vaccine information session on the willingness to get vaccinated among HCWs. A 45-min information session on the COVID-19 vaccine and vaccine hesitancy, followed by a question-and-answer session, was delivered by intensivists to 61 non-medical HCWs. The participants were questioned about their vaccine willingness before and after the information session: the acceptance rates increased drastically from 56% to 82%. Both of the papers address the importance of appropriate interventions that inform HCWs and facilitate exchange with HCWs to improve their vaccination intent. 

Another way of improving the vaccination rates among HCWs worldwide could be to make vaccination mandatory, either for all adults or just HCWs, as is already the case in some countries [53]. Only one paper advocated for mandatory vaccination and based this solely on the low vaccination intentions of certain respondents, but not on ethical argumentation [41]. The question arises if mandatory vaccination will solve the problem of low vaccination or if this would not merely increase distrust in vaccine-hesitant HCWs. Instead, Gur-Arie and colleagues [24] argued for improving trust between HCWs, health institutions and governments in order to promote voluntary vaccination. To force individuals to vaccinate for COVID-19 when their reasons not to vaccinate are due to safety concerns of the vaccine itself could have the opposite effect and may even lead to essential staff quitting their jobs. Therefore, addressing the concerns raised would be a more suitable way to build trust. One paper investigated which initiatives would increase vaccination intentions for COVID-19 among their HCW respondents [26]. These were: “increasing information quality about the vaccine (40%)”, “increasing information quality about vaccine development (28%)”, “implementing an economic incentive (21%)”, and “making vaccination mandatory (11%)”. Clearly, mandatory vaccination was the least-appealing option. Several papers analysed in this review have mentioned increasing information quality about the vaccine and its development [28,30,34,35,38,42,47]. Information quality can be increased through lectures, information sessions, focus groups or online interactive meetings. Increasing information quality can also be accomplished by spreading more informative videos on social media, where the most misinformation is spread and where the vaccine-hesitant individuals get information [28,34,38,47]. It seems that mandatory vaccination would not be ethically justifiable until all of the other means have been tried and failed.

### 4.3. Strengths and Limitations

The strengths of this paper include a large number of HCWs in the selected papers. The number of participants was 49,775 in all of the papers, of which 43,199 were HCWs, spread out over 16 different countries worldwide. An additional strength was the inclusion of both quantitative and qualitative papers. This resulted in both statistically comparable figures and more in-depth insights into the attitudes toward vaccination hesitancy worldwide. A limitation of this scoping review is the small number of papers on the attitudes of vaccine hesitancy of HCWs caring for people with ID. An alternative would have been to broaden the search to long-term care for the elderly in addition to people with ID, as HCWs in these facilities generally share characteristics in terms of the type and level of education, organization of care and focus on the quality of life. More research focussing on HCW in the field of ID are recommended.

The vaccine acceptance rates of HCWs in South America and Australia were missing in this scoping review. Additionally, a limitation of this review is that very few surveys were found with vaccine acceptance rates of HCWs from the African and Asian continents. In the systematic review by Sallam [13], these four continents were also under-represented. More research is necessary to investigate the vaccine intentions of HCWs across the globe to attain more insights into cultural differences, mainly in continents such as Africa, Asia, Australia and South America.

In the results section (Section 3), of this review, we confined ourselves to describing the socio-demographical predictors of vaccine willingness that were statistically significant in at least five papers. Thus, the predictors that were not described are: comorbidities; education; ethnicity; mental well-being; COVID self-history; geographical location; COVID family history; income and political orientation. Although these predictors were not very often significant, some of these were not questioned in the selected papers. These predictors might be statistically significant more often if they were investigated thoroughly. In order to examine the predictors of vaccine willingness, future research should include all of the possible parameters, such as mental well-being, COVID-19 family history and political orientation, as these were statistically significant most often in relative terms.

Furthermore, it was challenging to compare HCWs to the general population as this was not possible to retrieve from the papers. Related to this, for the predictors, it could be counted how often each factor was statistically significant. For the attitudes, this was not always the case. Most of the papers merely calculated the frequency of occurrence of the attitudes related to vaccine hesitancy. Finally, this paper reviews the papers that were found in April 2021. Given the variable nature of vaccine intentions, the development of different COVID-19 variants and the worldwide ‘booster’ vaccination programmes, new research should be executed quickly and continuously.

## 5. Conclusions

In summary, it appears that vaccine hesitancy varies drastically among HCWs worldwide, is driven by trust-related issues and is affected by the contextual factors. In order to reach the highest degree of vaccination, it is vital to address the problem of COVID-19 vaccine hesitancy more effectively, in the form of interactive interventions (e.g., meetings, webinars, discussion groups) concerning the trust-related issues of the vaccination programs, that address HCWs’ concerns and questions by providing personal stories and experiences. Furthermore, reliable and accurate information about the COVID-19 vaccination should be spread across social media platforms to target hard-to-reach population groups and to decrease misinformation. Additional research is needed to specify more precisely the attitudes concerning the vaccination intentions of HCWs globally and of the HCWs caring for people with ID.

## Figures and Tables

**Figure 1 ijerph-19-10192-f001:**
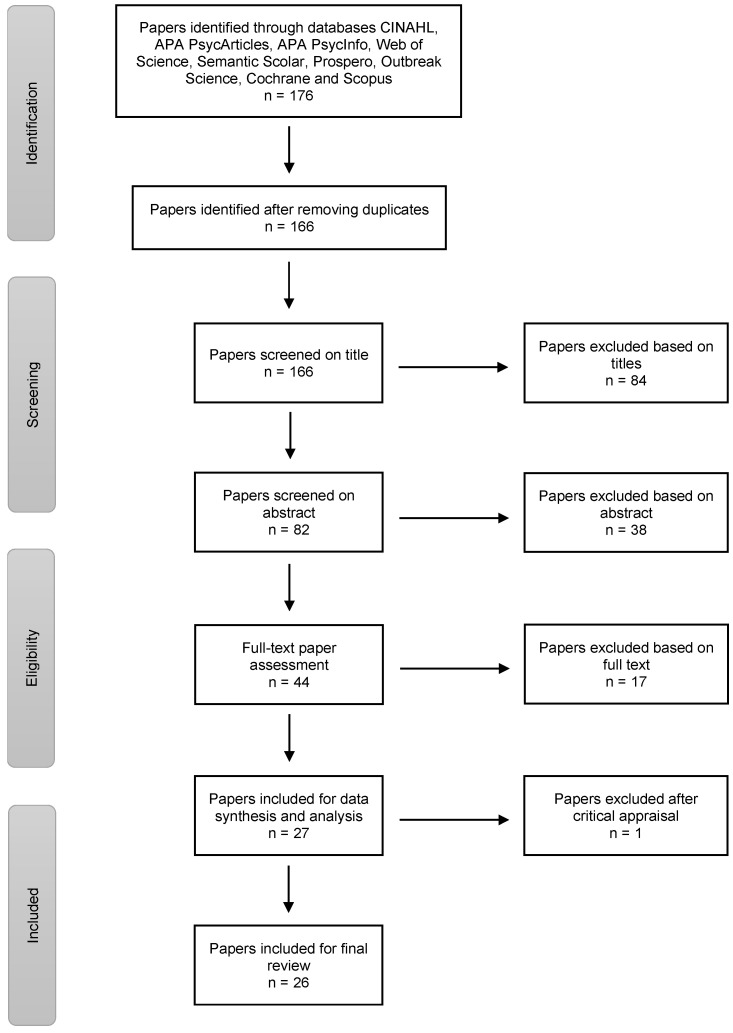
Flowchart of identification and selection process.

**Table 1 ijerph-19-10192-t001:** Characteristics of selected papers.

No.	Reference	Type of Research	Country of Sample	Continent	Date of Sample	Respondents (*N*)	Willing *	Undecided *	Hesitant *
1	Berry et al., 2021 [25]	Cross-sectional Qualitative Report	USA	North America	December 2020 to January 2021	HCW	n/a		
2	Di Gennaro et al., 2021 [26]	Cross-sectional Survey	Italy	Europe	October to November 2020	HCW Total (1723)	67.0%	26.0%	7.0%
						Specialised Medical Doctor (337)	69.0%		
						Medical Resident (259)	79.0%		
						Medical Doctor (544)	76.0%		
						General Practitioner (135)	63.0%		
						GP Trainee (70)	77.0%		
						Non-MD health professional (378)	43.0%		
3	Dror et al., 2020 [27]	Cross-sectional Survey	Israel	Middle East	March to April 2020	Doctors (338)	78.0%		
						Nurses (211)	62.0%		
						General Population Total (1112)	75.0%		
4	Eguia et al., 2021 [28]	Cross-sectional Survey	Spain	Europe	September to November 2020	HCW and General Population Total (731)	77.56%		
						Medicine (274)	82.5%		
						Nursing (51)	65.4%		
						Other HCW (37)	68.5%		
						Non-HCW (166)	76.1%		
						Unemployed (39)	79.6%		
5	Gagneux-Brunon et al., 2021 [11]	Cross-sectional Survey	France	Europe	March to July 2020	HCW Total (2047)	76.9%		
						Physicians (431)	92.1%		
						Pharmacists (501)	88.8%		
						Nurses (371)	64.7%		
						Assistant nurses (218)	60.1%		
						Midwives (37)	70.3%		
						Physiotherapists (24)	95.8%		
						Other (465)	67.1%		
6	Gakuba et al., 2021 [29]	Cross-sectional Survey	France	Europe	February 2021	HCW (61)	56.0%		
					February 2021 α	HCW (61)	82.0%		
7	Gur-Arie et al., 2021 [24]	Commentary	N/A	N/A	n/a	HCW	n/a		
8	Iadarola et al., 2022 [30]	Cross-sectional Survey	USA	North America	January to February 2021	HCW ID (258)	76.0%		
						Person with ID (91)	83.5%		
						Family ID (358)	73.5%		
						Combi HCW and Family (91)	80.2%		
						Other (27)	74.1%		
9	Kabamba Nzaji et al., 2020 [31]	Cross-sectional Survey	DRC	Africa	March to April 2020	HCW Total (613)	27.7%		
						Doctors (167)	37.7%		
						Nurses and other HCWs (446)	24.0%		
10	Kasozi et al., 2021 [32]	Cross-sectional Survey	Uganda	Africa	September to October 2020	HCW (260)	n/a		
11	Kose et al., 2020 [33]	Cross-sectional Survey	Turkey	Middle East	September 2020	HCW Total (1138)	68.6%	19.9%	11.4%
						Physicians (53)	50.9%	30.2%	18.9%
						Nurse/Midwife (306)	65.4%	20.9%	13.7%
						Student (Medicine and Nurse) (694)	72.3%	18.0%	9.7%
						Other (80)	61.3%	26.3%	12.5%
12	Kwok et al., 2021 [34]	Cross-sectional Survey	China	Asia	March to April 2020	HCW Nurses (1205)	63.0%		
13	Lunsky et al., 2021 [35]	Cross-sectional Survey	Canada	North America	January to February 2021	HCW ID (3371)	82.0%		
14	Manby et al., 2021 µ [36]	Cross-sectional Qualitative Report	UK	Europe	December 2020 to March 2021	HCW (24), Policies (24)	n/a		
15	Meyer et al., 2021 [37]	Cross-sectional Survey	USA	North America	December 2020	HCW (16,158)	55.0%	28.5%	16.4%
16	Mohamed-Hussein et al., 2021 µ [38]	Cross-sectional Survey	Egypt	Middle East	December 2020 to January 2021	HCW (496)	45.9%	13.2%	40.9%
17	O’Brien et al., 2021 µ [39]	Cross-sectional Survey	USA	North America	October 2020	HCW Total (2070)	54.2%		
						Paramedic/EMT (1449)	68.9%		
						Physicians (467)	64.0%		
						Other Practitioner (206)	53.9%		
						PA/NP (145)	49.7%		
						Nurse (586)	46.6%		
					December 2020 β	HCW Total (1541)	76.2%		
						Paramedic/EMT (1043)	75.0%		
						Physicians (318)	90.5%		
						Other Practitioner (149)	73.8%		
						PA/NP (106)	78.3%		
						Nurse (430)	66.9%		
18	Oliver et al., 2022 [40]	Cross-sectional Survey	USA	North America	December 2020 to February 2021	HCW Total (1933)	81% got. 11% plan to get.		
						Physicians (268)	95.0%		
						Research and Education (421)	92.0%		
						Advanced Practice Providers (87)	92.0%		
						Other Health Professionals (448)	78.0%		
						Nursing (265)	72.0%		
						Environmental Services (31)	58.0%		
						Administration, Logistics, Management (399)	71.0%		
						Community-based Providers (14)	29.0%		
19	Pădureanu et al., 2020 [41]	Cross-sectional Survey	Romania	Europe	April to May 2020	HCW Total (529)	69.0%		
						Physicians (344)	70.0%		
						Pharmacists (84)	53.0%		
						Nurses (31)	61.0%		
						Medical Students (66)	77.0%		
20	Qattan et al., 2021 [42]	Cross-sectional Survey	Saudi Arabia	Middle East	December 2020	HCW (673)	50.52%		
21	Sallam, 2021 [13]	Systematic Review	Worldwide	33 countries worldwide	February to December 2020	HCW (8 surveys), General Population (47 surveys), Parents/Guardians (3 surveys), UNI students (2)	n/a		
22	Shekhar et al., 2021 [43]	Cross-sectional Survey	USA	North America	October to November 2020	HCW Total (3479)	36.0%	56.0%	8.0%
						Direct Patient Care Providers (1573)	27.0%	62.0%	12.0%
						Direct Medical Providers (1207)	49.0%	48.0%	2.5%
						Administrative Staff (295)	34.0%	58.0%	8.5%
						Others without Direct Patient Contact (404)	45.0%	57.0%	9.2%
23	Szmyd et al., 2021 [44]	Cross-sectional Survey	Poland	Europe	December 2020	Medical Students MS (687)	91.99%	3.93%	4.08%
						Non-medical Students NMS (1284)	59.42%	18.85%	21.73%
24	Szmyd et al., 2021 [45]	Cross-sectional Survey	Poland	Europe	December 2020 to January 2021	General Population Total (1913)	54.31%	19.86%	45.64%
						HCW Total (387).	82.95%	5.94%	17.05%
						Medical Doctors (MD) (252)	94.44%	1.19%	5.56%
						Healthcare Assistants (HA) (135)	61.48%	14.81%	38.52%
25	Verger et al., 2021 [46]	Cross-sectional Survey	Belgium, France and Canada	Europe and North America	October and November 2020	HCW Total (2678)	72.4%		
			Belgium	Europe		HCW Belgium (414)	76.03%		
			France			HCW France (1209)	75.36%		
			Canada	North America		HCW Canada (1055)	70.41%		
26	Yurttas et al., 2021 [47]	Cross-sectional Survey	Turkey	Middle East	January 2021	HCW Total (320)	52.5%	26.6%	20.9%
						Physicians (152)	68.4%	20.4%	11.2%
						Nurses and medical/non-medical personnel (168)	38.1%	32.1%	29.8%
						Patients with RD Total (732)	34.6%	42.1%	23.3%
						General Population Total (763)	29.2%	51.8%	19.0%

DRC = The Democratic Republic of the Congo. Papers with multiple rows are banded according to respondents’ sample range: dark rows are total samples; light rows are sub-samples. Dotted lines mark a sample range change within the paper: total sample row or sub-sample row. * Percentage is only noted if reported in the paper; µ Paper is a pre-print; α Repeated measure with the same sample after an intervention. β Repeated measure with smaller sample after two months.

**Table 2 ijerph-19-10192-t002:** Socio-demographical predictors of vaccine willingness.

	Berry et al., 2021 [25]	Di Gennaro et al., 2021 [26]	Dror et al., 2020 [27]	Eguia et al., 2021 [28]	Gagneux-Brunon et al., 2020 [11]	Gakuba et al., 2021 [29]	Gur-Arie et al., 2021 [24]	Iadarola et al., 2022 [30]	Kabamba Nzaji et al., 2020 [31]	Kasozi et al., 2021 [32]	Kose et al., 2020 [33]	Kwok et al., 2021 [34]	Lunsky et al., 2021 [35]	Manby et al., 2021 [36]	Meyer et al., 2021 [37]	Mohamed-Hussein et al., 2021 [38]	O’Brien et al., 2021 [39]	Oliver et al., 2021 [40]	Pădureanu et al., 2020 [41]	Qattan et al., 2021 [42]	Sallam, 2021 [13]	Shekhar et al., 2021 [43]	Szmyd et al., 2021 [44]	Szmyd et al., 2021 [45]	Verger et al., 2021 [46]	Yurttas et al., 2021 [47]		
	1	2	3	4	5	6	7	8	9	10	11	12	13	14	15	16	17	18	19	20	21	22	23	24	25	26	N= *	N= ●
Profession		● *	● *	● *	● *	●		●	● *	●	● *		●		●	●	● *	● *	● *			● *		● *		● *	12	18
Age		● *	●	●	●	●		● *	● *	●	● *	● *	● *			● *	●	● *	●	●		● *	●	●	●	● *	10	21
Past vaccine behaviour		● *	● *		● *						● *	●		●		●		● *		● *		● *	● *	● *	● *		10	13
Gender		●	● *	●	● *	●		●	● *	●	● *	●	●			●	●	● *	●	● *		● *	●	●	●	● *	8	21
Comorbidities	●	● *		● *	●						●	●				● *				●		●				●	3	10
Education										● *			●			●				●		● *	●			●	2	7
Ethnicity								●					●	●	●		●	● *				● *					2	7
Mental well-being																							● *	● *		●	2	3
COVID self-history				●												●		●		●						● *	1	5
Geographical		●	●					●												●	●	● *					1	6
COVID family history		● *																									1	1
Income																●				●		● *		●			1	4
Political orientation																						● *					1	1

● item was asked/questioned. * Relation to vaccine willingness is statistically significant.

**Table 3 ijerph-19-10192-t003:** Attitudes toward vaccine willingness and vaccine hesitancy.

			Berry et al., 2021 [25]	Di Gennaro et al., 2021 [26]	Dror et al.,2020 [27]	Eguia et al., 2021 [28]	Gagneux-Brunon et al., 2020 [11]	Gakuba et al., 2021 [29]	Gur-Arie et al., 2021 [24]	Iadarola et al., 2022 [30]	Kabamba Nzaji et al., 2020 [31]	Kasozi et al., 2021 [32]	Kose et al., 2020 [33]	Kwok et al., 2021 [34]	Lunsky et al., 2021 [35]	Manby et al., 2021 [36]	Meyer et al., 2021 [37]	Mohamed-Hussein et al., 2021 [38]	O’Brien et al., 2021 [39]	Oliver et al., 2021 [40]	Pădureanu et al., 2020 [41]	Qattan et al., 2021 [42]	Sallam, 2021 [13]	Shekhar et al., 2021 [43]	Szmyd et al., 2021 [44]	Szmyd et al., 2021 [45]	Verger et al., 2021 [46]	Yurttas et al., 2021 [47]	
	Theme	Attitudes	1	2	3	4	5	6	7	8	9	10	11	12	13	14	15	16	17	18	19	20	21	22	23	24	25	26	N
Attitudes on Willingness	Health factors	COVID-19 Threat		●	●		● *							● *	● *			●		●	● *	● *		● *	● *	● *	●	●	14
	Ethics	Protect family members													● *					● *					● *	● *			4
		Mandatory vaccination																				● *		● *				●	3
		Collective responsibility												● *															1
		Vaccination is part of the job													● *														1
Attitudes on Hesitancy	Trust-related issues	Side effects	●	● ^1^	●	●		●		● ^1^			● ^1^	●	● *	●	●			●		●^1^		●	● *	● *		●	17
		Safety	●	●	●	●				●					● *	●	● ^1^	●		● *		●		●			● *	●	14
		Efficacy	●	●	●	● ^1^		● ^1^			● *			●		●		●		●		●		●				● ^1^	13
		Speed of development	●		● ^1^					●		●			● *	●	●	●		●		●		● ^1^			● *		12
		Distrust	●	●						●		●	●	●						●				●			● *	●	10
		Anti-vax and conspiracy	●	●							●							●		●					●	●		●	8
		Distrust because minority/ethnicity	●							● *		●				●				●									5
		Fertility	●													●				●									3
	Information	Lack of information/misinformation	●	●							●		●			●		●		●						●		●	9
	Practical factors	Alternative treatments									●	●																●	3
		Impact on other precautions	●																										1
		Logistics to get vaccinated													●														1
		Spirituality/religion	●																										1

● item was asked/questioned. * Relation to vaccine willingness or hesitancy is statistically significant. ^1^ Attitude is mentioned most often in relation to vaccine hesitancy.

## Data Availability

Not applicable.

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
