# Peer review of "COVID-19 Vaccination Intentions amongst Healthcare Workers: A Scoping Review"

_ijerph, 2022, doi:10.3390/ijerph191610192_

Round 1

Reviewer 1 Report

The authors have fully answered to my requests. The article is definitely valid and ready for publication

Author Response

Dear reviewer 1, 

Thank you for your positive review.

Yours sincerely,

The authors

Reviewer 2 Report

1. Please organise your work.It doesn t have to be so long, it might be half lengh, but organised.

2. Please learn to make references to any important information you are making along the text. The are so many examples. 

3. Why mix the two categories: HCW and HCW for ID? You only took into consideration two papers as a source. Not enough. Split the ideas and organise them. 

4. The precise comments are many, examples are many, so I prefer to suggest you another view of this review, because I can see you have make a big part of the work, but for the common reader it is too unclear, too fuzzy, and not very well refered to the literature quoted.

In addition, some examples.

AS a main observation: THe authors have included only two(2) papers for the idea of HCW/ID. I suggest to remake the review with only one focus, on the hesitancy of HCW.It needs more literature to include this other category of HCW.

Introduction: phrase between rows 38-44 needs to be refered to bibliography.There are So many afirmations that are not corelated to the literature.

section 4.1 rows 390, 391- please refer to the articles where the USA elections can interfere with the vaccine hesitancy. Some important afirmations have to be endorsed with references...

row 468, 474: the authors have to quote the five papers mentioned. If they want to publish in a journal(IJERPH) that requires much more precision in regard to the references.

rows 483-492: I found not fit, not relevant and rather childish , as we do not have to explain the definitions of variant forms of review when makimg altready a choice of a specific one. At least, please accept my opinion.

Conclusions: please concentrate 2-3 main ideas, with no more then one prospective study direction. It still needs to be organised the material.

A review, even a scoping review, cannot have only 41 references.I kindly suggest to the authors to take more time and take a zoom out view, so they can include a wider view of the matter.

Reviewer 3 Report

Since the manuscript has been resubmitted, it appears to be essentially a new work (removed from the title and main purpose of research on healthcare providers caring for patients with intellectual disabilities). However, despite the fact that the title was changed, a number of sections remained unchanged, making it difficult for those who were not familiar with the original version to read the article.

So, in Section 2.1. we meet "This scoping review aims to identify key themes in the literature concerning vaccination hesitancy and willingness among HCWs who care for people with ID."

The same is said in Section 2.3.

It is necessary to bring these sections in line with the new title and objectives of the work.

Author Response

This manuscript is a resubmission of an earlier submission. The following is a list of the peer review reports and author responses from that submission.

Round 1

Reviewer 1 Report

In this paper, the authors realized a literature review about vaccination intentions among healthcare workers (HCWs).

In total, 26 papers were identified concerning the vaccine intentions of 43,199 HCWs worldwide. Besides vaccine intentions, even predictors of and hesitancy, sources of vaccination information, contextual factors and changes over time were investigated.

In general, the article is methodologically well done, the problem is well introduced and the results are interesting. However, there are some points that authors should review.

FIRST

My main concern is about the topic of intellectual disability. While the authors would like to focus mainly on this aspect, so as to dedicate the title of the article to this topic, only 2 of the 26 articles take this issue into consideration. In fact, the article is more concerned with HCWs in general.

Although, of course, the problem is definitely relevant, it is necessary to dedicate the right space to this analysis: 1) reviewing the title, in which I would eliminate "in the field of intellectual disability"; 2) dedicating a specific subsection in the introduction and results.

SECOND

“According to Randolph and Barreiro [8], group immunity is achieved when 67% of the population is vaccinated, but other research states that group immunity thresholds are difficult to determine”

I would say that after 2 years of studies and field data, we have more up-to-date information on herd immunity. I would ask the authors to modify this sentence accordingly.

THIRD

Table 3 does not provide information about significance. Please, add this information.

Reviewer 2 Report

Overall, the idea is very actual and of interest.

I suggest therefore, in order to benefit from exposing the idea, to re-structure more clear the content: starting with the abstract.

Abstract: Please make it shorter, brief, compact. Try to find a clear+ wise way to communicate briefly and eloquent the conclusions.There is too much writing. 

Kew words: same observation

Introduction: in a word, please follow the previous instructions. And references should be more accurate, do not contract 3-4 items for one single phrase, while other paraghaphs do not have any.

Material and methods: eligibilty criteria: " Pre-printed and published qualitative, quantitative and commentary papers were included." - I do not agree with this criteria.

Item 2.3 has no objective, it is a description that is redundant, on my opinion.9 rows 111-136)

Results: the whole content is rather infantile than scientific: " First, the results section will describe the general characteristics of the literature. Then, 173

a more precise description will follow of the vaccination intention across samples in the literature. Next, the predictors of vaccine willingness will be discussed: the socio-demo- graphical characteristics of respondents. Finally, the attitudes on vaccine willingness and hesitancy will be discussed."

In this section, we expect results and not the a prospective plan...

Pages 6,7,8,9,10 are not results, but supplementary material. It is expected to interpretate the results.

Ref. to rows 228-244 - wrong section, they belong maybe to methods, not to results.

My overall conclusion: There are  confusions of the structure of the article, and a lengh that is not justified through its content. Large amount of text that makes the article difficult to read and more difficult to understand.

Reviewer 3 Report

Dear authors,

Thank you for the interesting study.

I think it will be of great interest to the readers. I have only one minor concern. Change the title of the manuscript. You have written that “only two papers reported on HCWs in the care of people with intellectual disabilities. In both articles, the HCWs showed a comparable percentage of vaccination intention; 76% and 82%.”. Your study discusses a wider range, evaluating all healthcare professionals.

Therefore, I propose to make appropriate changes to the Title, Introduction and Discussion parts of the manuscript. This will not require much effort, but then these parts will fully reflect the essence of the work.
